# Biofortified Crops for Combating Hidden Hunger in South Africa: Availability, Acceptability, Micronutrient Retention and Bioavailability

**DOI:** 10.3390/foods9060815

**Published:** 2020-06-21

**Authors:** Muthulisi Siwela, Kirthee Pillay, Laurencia Govender, Shenelle Lottering, Fhatuwani N. Mudau, Albert T. Modi, Tafadzwanashe Mabhaudhi

**Affiliations:** 1Dietetics and Human Nutrition, School of Agricultural, Earth and Environmental Sciences, University of KwaZulu-Natal, Private Bag X01, Scottsville 3209, Pietermaritzburg 3201, South Africa; siwelam@ukzn.ac.za (M.S.); pillayk@ukzn.ac.za (K.P.); govenderl3@ukzn.ac.za (L.G.); 2Centre for Transformative Agricultural and Food Systems, School of Agricultural, Earth and Environmental Sciences, University of KwaZulu-Natal, Private Bag X01, Scottsville 3209, Pietermaritzburg 3201, South Africa; sewells@ukzn.ac.za (S.L.); modiat@ukzn.ac.za (A.T.M.); 3School of Agricultural, Earth and Environmental Sciences, University of KwaZulu-Natal, Private Bag X01, Scottsville 3209, Pietermaritzburg 3201, South Africa; mudauf@ukzn.ac.za

**Keywords:** biofortification, hidden hunger, malnutrition, nutrient-dense crops

## Abstract

In many poorer parts of the world, biofortification is a strategy that increases the concentration of target nutrients in staple food crops, mainly by genetic manipulation, to alleviate prevalent nutrient deficiencies. We reviewed the (i) prevalence of vitamin A, iron (Fe) and zinc (Zn) deficiencies; (ii) availability of vitamin A, iron and Zn biofortified crops, and their acceptability in South Africa. The incidence of vitamin A and iron deficiency among children below five years old is 43.6% and 11%, respectively, while the risk of Zn deficiency is 45.3% among children aged 1 to 9 years. Despite several strategies being implemented to address the problem, including supplementation and commercial fortification, the prevalence of micronutrient deficiencies is still high. Biofortification has resulted in the large-scale availability of βcarotene-rich orange-fleshed sweet potatoes (OFSP), while provitamin A biofortified maize and Zn and/or iron biofortified common beans are at development stages. Agronomic biofortification is being investigated to enhance yields and concentrations of target nutrients in crops grown in agriculturally marginal environments. The consumer acceptability of OFSP and provitamin A biofortified maize were higher among children compared to adults. Accelerating the development of other biofortified staple crops to increase their availability, especially to the target population groups, is essential. Nutrition education should be integrated with community health programmes to improve the consumption of the biofortified crops, coupled with further research to develop suitable recipes/formulations for biofortified foods.

## 1. Introduction

Hidden hunger, or micronutrient deficiency, is a leading global problem of public health importance, especially in sub-Saharan Africa, the Caribbean, and East, South Eastern and Western Asia (Refer to Table 1) [1]. Therefore, the biofortification of staple crops with target micronutrients is essential, with the aim of curbing malnutrition and diseases and promoting the wellbeing of the target population groups. Among the micronutrients, vitamin A, iron and zinc (Zn) were identified as common deficiencies among economically disadvantaged communities. Thus, they pose a detrimental effect on the health, wellbeing and socio-economic upliftment of the affected population groups [2]. Moreover, vitamin A, iron and Zn deficiencies are associated with over 50% of all deaths of under-five year olds globally [3]. Vitamin A deficiency (VAD) is the main cause of preventable night blindness, childhood morbidity and mortality [4]. Iron deficiency (ID) is the leading cause of preventable iron deficiency anaemia, poor cognitive development, and maternal and childhood deaths [5]. In contrast, zinc deficiency (ZnD) is associated with childhood diarrhoea, impaired immunity and reduced linear growth [6,7]. Approximately 30% of children under the age of five years are stunted in sub-Saharan Africa [7]. Moreover, deficiencies of vitamin A, iron and Zn are associated with childhood stunting [8]. The consequences of stunting are also alarming. Stunted children have poor cognitive development [9,10] and are at risk of developing non-communicable diseases like cardiovascular disease, type 2 diabetes mellitus and obesity in later adult life [11,12,13,14].

The prevalence of VAD, ID and ZnD remain unacceptably high in some sub-Saharan African countries, including South Africa. A recent systematic review conducted in four sub-Saharan African countries (South Africa, Ethiopia, Nigeria and Kenya) revealed that the prevalence of ID anaemia ranged from 25–53%, ID from 12–29%, VAD from 14–42% and ZnD from 32–63% in persons aged 0 to 19 years [15]. The South African Government formulated and implemented various strategies to overcome micronutrient deficiencies. These included dietary diversification, supplementation and commercial food fortification [16,17]. However, these strategies have not adequately addressed micronutrient deficiencies because nutritional supplements, commercially fortified foods and diversified foods, including animal food sources, are unaffordable and inaccessible to the rural poor [18,19]. To this end, biofortification has been suggested as a feasible alternative.

Crop biofortification is a process involving increasing the concentration of target nutrients in staple food crops by genetic manipulation through conventional breeding and recombinant DNA (rDNA) technology [20]. South Africa is among the first African countries to adopt and promote the development of biofortified crops [21]. The biofortification of crops in Africa has targeted three problematic micronutrients, namely vitamin A, iron and Zn [22]. The main goal of biofortification is to ensure that high-quality biofortified varieties are available and easily accessible, both physically and economically, are acceptable to the target consumers and, when consumed, that the target nutrients are bioavailable [20,22].

There is a need to monitor the progress of the biofortification programme to inform policymakers on how to improve it and its implementation. This paper aimed to review the prevalence of micronutrient deficiencies (vitamin A, iron and Zn) in South Africa. Furthermore, this paper examined the availability of food crops biofortified with vitamin A, iron and/or Zn, and their acceptability to target consumers. Moreover, micronutrient preservation during the processing and preparation of biofortified foods, and the bioavailability of the target nutrients, was also assessed.

To address the aims of this paper, a mixed method review approach was utilized. Information on the prevalence of vitamin A, iron and zinc deficiencies was obtained from the report of the South African National Health and Nutrition Examination Survey (SANHHES-1) published in 2013. Relevant literature on the availability of biofortified food crops, their acceptability, nutrient retention during processing and the bioavailability of the target nutrients was acquired from the internet database using search engines, including ScienceDirect, Google Scholar and PubMed. To acquire relevant research studies for this review, the following keywords and Boolean operation combinations were used: micronutrient/s, vitamin/s, mineral/s, vitamin A, iron, zinc, deficiency/ies in South Africa; availability, biofortified, crop/s AND provitamin A, iron, zinc, maize, bean/s, rice, sorghum, millet, cassava, sweet potato; consumer, acceptability, acceptance OR preference, biofortified, food/s, crop/s, provitamin A, maize, bean/s, rice, sorghum, millet, cassava, sweet potato; processing, preparation, retention AND nutrient/s, iron, zinc, provitamin A, biofortified, food/s, crop/s, maize, bean/s, rice, sorghum, millet, cassava, sweet potato. The scientific literature was then complemented with documented information on South African Food and Nutrition Security Policies and Strategies, which was obtained from the National and Provincial Governments Gazettes of the Republic of South Africa.

**Table 1 foods-09-00815-t001:** Total population at risk of major micronutrient deficiencies and top five staple crops, by region [23].

	Asia	Africa	Latin America and the Caribbean	Total Cases of Deficiency/Inadequate Intake
**Total population at risk**				2,466,226,780
All	1,722,763,911	541,818,522	201,644,347	994,556,079
Iron	699,198,517	237,395,434	57,962,128	1,273,705,384
Zinc	901,336,413	236,801,679	135,567,293	197,965,317
Vitamin A	122,228,982	67,621,409	8,114,927	197,965,137
**Total Kilocalories per day (millions)**				
Rice	3,146,030	201,275	141,990	3,489,295
Wheat	2,017,353	358,305	194,579	2,570,236
Maize	301,673	352,693	211,579	866,175
Potatoes	223,633	34,527	24,846	283,007
Cassava	71,263	140,542	31,554	243,359

## 2. Micronutrient Deficiencies in South Africa

The biofortification of staple foods with vitamin A, iron and Zn is widely recognised in African countries, including South Africa, to alleviate hidden hunger [20]. South Africa was the first country in Africa to adopt the biofortification of staple food crops [21]. Moreover, it is argued that the biofortification of staple foods like sweet potato may contribute to food and nutrition security in South Africa [22,24]. Therefore, we must monitor the prevalence of these micronutrient deficiencies before and after the adoption of biofortification. Such findings can be essential to inform policymakers about the impact of the biofortification program and devise ways to improve its delivery.

### 2.1. Vitamin A Deficiency

The prevalence of VAD (serum retinol < 20 µg/dl) among South African children below five years old increased from 33% in 1994 [25] to 43.6% in 2012 (Figure 1) [26]. This could be attributed to the consumption of foods devoid of vitamin A, as evidenced in the National Food Consumption Survey (NFCS) of 1999, which reported that 50% of children had a vitamin A intake of less than half the Recommended Dietary Allowance (RDA) [27]. The RDA is defined as the average daily dietary intake level that is sufficient to meet the nutrient requirements of nearly all healthy individuals in a group [28]. Furthermore, the 2005 National Food Consumption Survey (NFCS-FB) indicated that over 63% of South African children between the ages of 1 and 9 years were found to be vitamin A deficient [29,30]. The health risk for children with VAD is such a concern that the National Department of Health created a vitamin A supplementation (VAS) policy in 2012 for children below five years old [30].

The prevalence of VAD in South Africa makes a compelling case for the further and continuous implementation of proven interventions like vitamin A supplementation, fortification and provitamin A carotenoid biofortification. A school feeding intervention programme that involved fortifying biscuits with ꞵ-carotene and other micronutrients, targeting school-going children between 6 to 11 years of age in rural KwaZulu-Natal, was monitored for 30 months to assess its impact on vitamin A status. Each child received three shortbread-based biscuits providing 50% of the ꞵ-carotene RDA for 7–11 year old children (2.1 mg ꞵ-carotene) [31]. A substantial improvement in serum retinol was reported; however, when the school reopened after the summer holidays, serum retinol returned to pre-intervention levels [31]. This suggests that the children were not consuming vitamin A-rich foods while they were at home during the holidays. Home gardening could provide a complementary solution to alleviating VAD, because it has been demonstrated that, when caregivers grow biofortified crops, they feed children with biofortified foods, resulting in significant improvements in serum retinol levels among children from rural South Africa [32].

### 2.2. Iron Deficiency

Children have a higher risk of suffering from iron deficiency (ID), defined as serum ferritin <1.5 µg/dL [33]. Two larger South African national surveys have shown that the prevalence of ID among children below five years old increased from 10% in 1994 [25] to 11% in 2013 [26]. In contrast, the prevalence of iron deficiency anaemia (IDA) decreased from 5% in 1994 [25] to 2.1% in 2013 [26] (Figure 1). It is argued that the decrease in the IDA prevalence can be partly explained by the national food fortification programme legislated in 2003 [26]. In South Africa, despite the legislated fortification of staple foods (National Food Fortification 2002), there is a shortage of iron-fortified foods in the diets of most children, leading to a low dietary intake of iron [26].

A high prevalence of ID (20.9%) was reported among primary school-going children in KwaZulu-Natal province [34]. To this end, micronutrient fortification through the school feeding programme was introduced [31]. Stuijvenberg and colleagues evaluated the long-term effect of the programme on the micronutrient status of primary school children aged 6 to 11 years, who were given a ꞵ-carotene-, iron- and iodine-fortified biscuit [31]. Children were followed in longitudinal study surveys which were performed in the same school for 2.5 years. Findings revealed that there was a significant improvement in the indicators of iron status, such as serum ferritin, haemoglobin and transferrin saturation [31]. However, after the summer holidays, all the iron status indicators returned to pre-intervention levels. This finding suggests that other household-level solutions, like growing iron-biofortified food crops at home for household food and nutrition security, are paramount and may have a more sustainable impact [22].

### 2.3. Zinc Deficiency

A national survey, conducted in 1999, revealed that 32–53% of South African children aged 1 to 9 had a Zn intake that was below the RDA [27]. A follow-up survey, the South African National Food Consumption Survey Fortification Baseline (NFCS-FB-1) (2005), showed that 45.3% of children in the same age group had inadequate Zn intakes (below the RDA) [29]. These findings are a public health concern because a prevalence of insufficient Zn intake of more than 25% is considered an elevated risk for population ZnD [35]. The RDA for Zn for children aged 1 to 13 years old is 3, 4 and 8 mg/day for children aged 1–3, 4–8 and 9–13 years old, respectively [36].

The two studies did not identify ZnD in the population using Zn biomarkers, probably because Zn is widely distributed in the body with no specific stores [37]. However, they assessed for Zn status using the dietary intake of Zn. The assessment of dietary Zn is considered an adequate proxy measure of Zn status in a population [38,39].

## 3. Feeding Practices and Micronutrient Deficiencies in South Africa

Micronutrient deficiencies, such as vitamin A, iron and Zn, are prevalent among children in South Africa. This is particularly due to the consumption of a diet inadequate of these micronutrients [27,40]. The most recent South African National Health and Nutrition Examination Survey (SANHANES-1) revealed that the mean Dietary Diversity Score (DDS) for all age categories was at a national level of 4.2 [26]. However, there was some disproportion of DDS based on province and race. For example, the Western Cape (28.2%) and Gauteng (26.3%) reported the lowest number of participants with low DDS or DDS < 4, while the North West (61.3%) and Limpopo (65.6%) had the highest number with low DDS [26]. The average nutritional diversity score was higher among white participants compared to other race groups. In contrast, the lowest nutritional score and highest number of participants with low dietary diversity were black African participants (44.9%) [26]. This disproportion in DDS based on province and race indicates that nutrition programmes should identify and target the most disadvantaged populations, considering both demographics and geography.

In South Africa, supplementation and fortification strategies have been adequately implemented [41]. These strategies appear to be sustainable, but supplementation has been ineffective in reaching children aged 12–59 months, as they are not routinely taken to health care facilities for immunisation after the age of 18 months [30]. Additionally, fortified foods have become unaffordable for those in rural communities [41,42]. Moreover, an evaluation of vitamin A supplementation programmes in several developing countries indicates that vitamin A supplementation alone does not prevent VAD [43]. Policymakers should consider complementing the already existing strategies with new ones, such as biofortification.

## 4. Crop Biofortification as a Strategy

Plant breeding and modern biotechnology are crucial in increasing the micronutrient density of staple crops. Biofortification has the potential to improve the nutritional status and health of poor populations in rural and urban areas in developing countries [23]. The common approaches to biofortification are conventional breeding, recombinant DNA (rDNA) technology [23,44] and agronomic management [45]. However, conventional breeding is currently at its early stage of research and development. Traditional plant breeding includes finding and developing parent lines with a naturally high concentration of the target nutrient and crossing them over time to produce the desired concentrations of the nutrient and agronomic traits in the plants, whereas rFDNA technology increases concentrations of the target nutrient in a crop by inserting genes from another species to produce transgenic crops [44]. Agronomic biofortification offers temporary increases in micronutrients through agronomic management methods, such as fertilizers and/or foliar sprays, which are particularly effective if the target micronutrients are absorbed by the plant directly from the soil [45].

### 4.1. Biofortification by Conventional Breeding and Transgenics (rDNA Technology)

The National Policy on Food and Nutrition Security for the Republic of South Africa supports biofortification through the Household Food and Nutrition Strategy, that recognises measures, including biofortification of staples, to combat malnutrition [46]. Furthermore, the Agricultural Research Council (ARC), a South African Government agricultural research agency, runs a programme on plant breeding. The Plant Breeding Programme specialises in research on the breeding of indigenous food crops, including the biofortification of OFSP with provitamin A carotenoids and research on the biofortification of maize to contains provitamin A carotenoids.

HarvestPlus, a research organisation programme implemented by the ARC with international research institutes, targets a variety of crops that are part of the staple-based diets of the rural and urban poor, and breeds them to ensure that they are rich in Fe, Zn and provitamin A [22]. HarvestPlus formed a multidisciplinary alliance of experts at various institutions globally, including South Africa [20]. Table 2 shows the staple food crops that HarvestPlus started with more than a decade ago, with their specific micronutrients, agronomic traits and target country [22,47].

From 2004 to 2017, the HarvestPlus programme, in conjunction with the Consultative Group on International Agricultural Research (CGIAR), increased the release of biofortified varieties of food crops every year, as shown in Figure 2 [48]. Furthermore, by the year 2018, HarvestPlus, working in collaboration with partners, was undertaking additional testing of crop biofortification in 30 countries where biofortification had not yet been released [48]. The CGIAR argues that these activities will lead to the additional release of 12 staple crop varieties, meeting farmers’ demands for increased quality, yield and climate resilience in these counties [48].

The transgenic approach is not the most common approach for biofortification, as it is time consuming, expensive and not adopted by a large number of people [49]. The efficacy of transgenic biofortification has been confirmed for the majority of major staple crops, including maize and rice, as well as crops targeted for their nutritional value [49,50]. Substantial genetic variation for functional traits exists among cereals and tuber crops. The provitamin A content in cassava accessions held in major gene banks ranges from 0–19 ppm [51]. A similar range of provitamin A content was reported in a global maize core collection [52]. Both classical and modern breeding techniques have significant effects on the improvement of essential mineral elements such as Zn and Fe, as well as provitamins [53]. A comparison of normal and transgenic crops showed that provitamin A levels could be increased by 1.6 to 37 µg/g DW in rice and maize when phytoene synthase (PSY) was used. Over expression of the ferritin gene, *Syfer H-1*, caused an increase of Zn up to 38 µg/g DW, and of Fe up to 35 µg/g DW. Bacterial *crtB* improved cassava vitamin A by 6.67 µg/g DW [53]. Beta carotene was found to increase by 7–13 fold when bacterial *CrtB* and *CrtI* genes were employed in transgenic maize [54]. Another study [55] showed that β-carotene increased by 169-fold in transgenic maize that was comprised of provitamin A, ascorbate and folate. Furthermore, [56] found increased levels of carotenoids in transgenic sweet potato cells that contained the IbCHY-β gene. The maximum carotenoid content found in the transgenic sweet potato was 117 mg/g DW [56]. Nonetheless, GMOs have astonishingly high nutritional contents compared to conventionally bred plants. For example, Cassava expressing the bacterial *CrtB* gene amassed up to 21 μg/g of carotenoids, a 34-fold increase when compared to the wild type [57]. Likewise, the total carotenoid content in a transgenic maize inbred line 642 was 5.7 mg kg^−1^ dry mass, compared to 3.1 mg kg^−1^ dry weight in conventional QPM [58].

Both contemporary and classical plant breeding strategies are useful in the biofortification of crops. In developing countries, plant breeders relay nutrient-dense genotypes and molecular markers to introgress quality traits into elite germplasm. Further genetic variation is created through mutagenesis, as is the case with the lysine-enriched maize *opque-2* mutant. However, a significant challenge associated with conventional breeding is the time taken to identify useful traits and breed them into improved cultivars. On the contrary, transgenic strategies offer a rapid way to introduce desirable traits into new varieties. Genetic engineering is predominantly practical when the nutritional element is synthesized *de novo* by the plant or available in the environment. In this regard, mechanisms regulating trait expression can be increased or suppressed. Despite the high genetic gains accrued through genetic engineering, the acceptability of transgenic crops remains a bone of contention.

### 4.2. Provitamin a Biofortified Crops in South Africa

Vitamin A can be attained from food in the form of preformed vitamin A in animal food sources, such as eggs, liver and dairy products, or provitamin A carotenoids, mainly β-carotene in plant products, such as green leafy and yellow-coloured vegetables [59]. The OFSP has been released as a provitamin A biofortified food crop in South Africa (Table 2). However, testing of the provitamin A-biofortified maize is still underway.

### 4.3. Orange-Fleshed Sweet Potato

Several cultivars of OFSP produced by conventional breeding and transgenic biofortification are available in South Africa for human consumption and continuous production through home gardening and large scale farming. Production of the OFSP is being supported by the Agricultural Research Council (ARC) Vegetable and Ornamental Plant Institute (ARC-VOPI) [60]. Breeding of OFSP in South Africa started in 1996, with sweet potatoes containing adequate concentrations of β-carotene [61].

Carrots and spinach are rich sources of ꞵ-carotene [62]. However, OFSP contains higher amounts of ꞵ-carotene than spinach and carrots. On average, the ꞵ-carotene content of carrot and spinach is 112.10 and 99.40 µg/g DW, respectively [62], compared to the South African OFSP varieties that range from 142.10 to 207.79 µg/g DW of OFSP [63]. Retinol, measured in Retinol Activity Equivalents (RAE) is an indicator of functional vitamin A in the body [64], with 12 µg β-carotene = 1 µg retinol = 1 RAE [28]. Therefore, in this case, carrot and spinach provide 934 and 828 RAE, respectively compared to OFSP, which provides higher amounts of retinol, ranging from 1184 to 1731 RAE. However, when consumed by humans, spinach and carrot improve vitamin A status at lower levels [65] compared to OFSP [66].

The Recommended Dietary Allowance (RDA) for vitamin A for a child aged 4–8 years old is 400 RAE [28,36]. A root (100 g) containing a medium intensity OFSP variety has the ability to meet the daily vitamin A needs of a child aged 4–8 years [67].

A study led by [68] to determine the concentration of trans-β-carotene and selected minerals in 12 varieties of sweet potatoes (*Ipomoea batatas*), produced at four agro-geographical production sites in South Africa had the potential to contribute 100% of the RDA for vitamin A in children aged 4–8 years old. The trans-β-carotene concentration of the varieties were wide-ranging across geographical areas. The authors argued that the variations within varieties across geographical areas were attributable to changes in soil mineral content, soil pH and the interaction of the aforementioned factors [68].

To confirm that households have access to OFSP planting material, a community-based sweet potato nursery was established in one of the South African rural areas in 2003 [69]. In the rural Western Cape province, OFSP production was scaled up, which improved community participation in the programme [70]. In rural South Africa, people accessed provitamin A carotenoid-rich foods from either the community or home garden [69]. However, the supply of OFSP to the households was the lowest, as shown in Figure 3 [69].

### 4.4. Provitamin A Biofortified Maize

The staple food for the vast majority of the rural and urban poor in South Africa is maize [26]. Maize is a well-established vehicle for provitamin A carotenoid biofortification. However, there is no biofortification programme for maize in South Africa (Figure 2). In contrast, there is commercial fortification of maize [71,72], which may not be affordable or sustainable for the rural and urban poor in South Africa. It is promising that the government of South Africa is promoting research on the biofortification of maize to implement it at a level that can alleviate VAD [73,74,75].

The ꞵ-carotene content of provitamin A-biofortified maize is similar to provitamin A-rich spinach and carrot [76]. However, maize is a richer source of the provitamin A carotenoid, β-Cryptoxanthin. Moreover, β-Cryptoxanthin from provitamin A-biofortified maize is highly bioavailable in humans [77], with 24 µg of β-cryptoxanthin = 1 µg Retinol = 1 RAE [28]. Titcomb and colleagues fed young adults a daily provitamin A-biofortified maize meal containing 500 µg β-Cryptoxanthin [77] and 500 µg of β-cryptoxanthin provided an additional 21 RAE to add onto the RAE contributed by β-carotene in provitamin A-biofortified maize [77]. The fat content of a typical provitamin A-rich carrot is lower than that of provitamin A-biofortified maize, as the fat content in carrots is less than 0.5% compared to 3–18% dry weight in provitamin A-biofortified maize [78]. The high fat content in maize biofortified with provitamin A is important in order to increase the absorption of provitamin A carotenoids, as they are fat-soluble [36].

### 4.5. Zinc and Iron Biofortified Crops

The dry bean, also referred to as the common bean (*Phaseolus vulgaris* L), is a yearly leguminous crop. The dry bean is suitable for biofortification with iron [79,80]. HarvestPlus has suggested the dry bean as one of the potential vehicles for Zn and iron biofortification [81]. The dry bean is a staple food for rural and urban poor dwellers in South Africa. Biofortification of the dry bean has been suggested as a complementary approach to addressing ZnD in South Africa [81]. However, there is low production of the dry bean through gardening in South Africa, as some of the beans are imported into the country from neighbouring Zimbabwe, Malawi and Zambia. Therefore, the consideration of the biofortification of locally-produced dry beans or adoption of existing biofortified dry beans is essential. The HarvestPlus program developed red-mottled dry beans, which are characterised by increased mineral content, increased yield, and tolerance to drought and diseases [81].

Generally, the dry bean grain contains 3.14–12.07 mg/g iron and <1.89–6.24 mg/g Zn [82], compared to the NUA genotypes, which contain 40–90 mg/kg Fe and 10–35 mg/kg Zn [81]. Following the release of the NUA dry beans, they were assessed under diverse field conditions in several geographic areas, which included Latin America, and eastern and southern Africa, forming part of the HarvestPlus and Agrosalud programmes [83]. Beans of NUA35 and NUA56 contain significantly higher concentrations of Zn than commercial cultivars commonly cultivated in Uganda, Rwanda, Colombia, Bolivia, Costa Rica and Guatemala, particularly when assessed under varying climates, altitudes and soil types [80]. The NUA dry bean genotypes have been assessed for disease and drought tolerance at the ARC Cedara and Makhathini Research Stations in South Africa [81]. South African institutions are testing these genotypes, indicating that the government aims to adopt iron and Zn-enhanced beans [80].

## 5. Agronomic Biofortification

Changing environmental conditions, including climate change and changes in soil composition, tend to impact negatively on agricultural production. This calls for a shift to agricultural practices that counter the negative impacts of the changing environmental conditions on agricultural production. Agronomic biofortification and agronomic management are being evaluated for wide adoption to combat the negative effects of changes in the environment on agricultural production. Agronomic biofortification offers a temporary micronutrient increase in the soil through fertilizers and/or foliar sprays, and is useful in increasing the micronutrients absorbed directly by the plant. A variation in the β-carotene content of OFSP grown in different geographic locations was found in South Africa. They ascribed these finding to variations in soil mineral content, soil pH and the interaction of these factors [68]. A study was conducted by [61] in South Africa to assess the agronomic ability, stability and genetic diversity of recently developed OFSP genotypes. This included the evaluation of twelve entries, nine of which had an orange flesh colour, at four sites for two seasons in multi-environment trials. The Cultivar Impilo developed stable, high root yields similar to the commercial cultivar Beauregard (the control), while the Purple Sunset had specific adaptability and increased yield. Both varieties showed attributes of suitable dry mass and satisfactory taste [61].

Other agronomic problems, such as drought, need to be considered in agronomic biofortification and management research. For example, South Africa is drought-prone [84], yet OFSP requires sufficient amounts of water for proper growth and high yields. Several projects that promoted the home production and daily intake of conventionally biofortified OFSP in South Africa were reviewed, which established that they improved the vitamin A status of vulnerable groups, including children under the age of five years [85].

A study conducted in Pretoria determined β-carotene content and β-carotene yield in incremental water and chemical fertilizer applications for OFSP in separate field trials [86]. β-carotene content was 14% higher for intermediate (50%) and high (100%) fertilizer treatments, in contrast to the 0% fertilizer treatment. On the other hand, β-carotene yield improved two-fold and four-fold in the intermediate and high fertilization treatment, respectively [86]. This finding may echo that conventional biofortification is essential, but not adequate, to meet the demands of biofortification; therefore, agronomic biofortification should not be overlooked. Moreover, several projects that promoted the home production and daily consumption of OFSP through agronomic biofortification have shown these factors to significantly improve the vitamin A status of their target population [24].

Drought is the single most critical yield-limiting factor in areas where cultivation depends on rainfall [84]. Therefore, irrigation and/or research to identify drought-tolerant cultivars are more critical in drought-prone areas, compared to areas receiving adequate rain. A study conducted by [87] screened accessions of sweet potato for drought resilience using a rapid technique; these were then monitored by field screening to identify accessions that performed optimally under varying degrees of water stress conditions in South Africa. Twelve of the best performing accessions were selected for field trials, which were conducted in Lwamondo, Limpopo province, an area of endemic drought in South Africa. The study found that Za-pallo, Tacna, Ejumula, 2004-9-2 and Ndou were the best performing accessions [87].

In addition [60,86] determined that (i) β-carotene content, yield and water efficiency at incremental water application, and (ii) β-carotene content and yield at incremental chemical fertilizer application, were evident for OFSP. Thus, β-carotene content ranged between 15 and 34% in the low irrigation treatment, which was higher than the optimal irrigation treatment [70]. An increase in water lead to a two-fold rise in β-carotene yield per unit area. The most suitable blend of β -carotene yield and water productivity were attained at the intermediate level (60%). The study revealed that 1 ha of OFSP produced, at the intermediate water application level, a yield level of 24.6–28.4 t ha^−1^, which could possibly provide 452–730 households (of six persons) with a suitable quantity of provitamin A over a 180 day period [86].

Marginal soils are commonly used to cultivate sweet potatoes, which consist of low agricultural inputs and can be harvested when required for consumption. A study [63] found that OFSP harvested at optimal cultivation conditions four, five and six months after planting, produced under rural settings, were smaller, had increased β-carotene content and needed a smaller serving size to offer 100% of the dietary vitamin A requirements in comparison to those generated under ideal settings. Harvesting at four, five and six months after planting showed a steady increase in β-carotene content at the rural village level. However, there was no change observed regarding harvest time under optimal conditions. The aforementioned factors should be accounted for when considering food-based interventions that are aimed at dealing with vitamin A deficiency, especially when grown in non-commercial settings.

### Zinc Agronomic Biofortification

South African soils are deficient in Zn and iron [88]. Soil Zn deficiencies can affect plants negatively through stunted growth, resulting in a decreased number of tillers, chlorosis, smaller leaves, longer maturity periods, spikelet sterility and the sub-standard quality of harvested products [89]. The high prevalence of Zn deficiency among the rural poor in South Africa is partly due to eating foods cultivated in soils with a low concentration of Zn [88].

Studies that have evaluated Zn agronomic biofortification have shown positive results. For example, Zn was applied in pots as ZnSO4·7H2O to the maize cultivar DK–6142 as a foliar spray (0.5% *w*/*v* Zn sprayed 25 days after sowing and 0.25% *w*/*v* at tasselling), surface broadcasting (16 kg Zn ha^−1^), subsurface banding (16 kg Zn ha^−1^ at a depth of 15 cm), surface broadcasting and foliar, and subsurface banding and foliar, in comparison to an unfertilized control [90]. All treatments yielded increased growth and nutritional attributes in maize when compared to the control. In addition, Zn fertilization reduced grain phytate significantly and improved grain Zn concentration [90]. The reduction of phytate seen in the maize grains in that study was significant because phytate is a key anti-nutrient factor found in staple food cereals such as maize [91].

In a separate study, the agronomic biofortification of wheat was evaluated across seven countries (China, India, Kazakhstan, Mexico, Pakistan, Turkey and Zambia). Foliar Zn application alone, or in combination with soil application, significantly increased Zn concentrations in wheat grain by 84% and 90%, respectively [92]. Furthermore, a recent review revealed that providing crops with adequate amounts of Zn through the soil and foliar fertilizer under field conditions was crucial for biofortification efforts for Zn [93].

## 6. Bioavailability of Target Micronutrients in Biofortified Foods

Bioavailability is the portion of a consumed nutrient that is accessible for utilisation in normal physiological functions and/or for storage [94]. Studies conducted in humans provide important information on bioavailability when the nutrient of interest is present in the blood stream (available for utilisation), in tissue (storage) and in excretions like urine [95]. The target nutrient in the biofortified food consumed should be bioavailable in order for bioavailability to be successful.

### 6.1. Provitamin A Bioavailability

Effective variations from provitamin A to retinol, a form of vitamin A used by the body, were found in vitamin A bioavailability studies. [96]. Additionally, efficacy studies established that increasing provitamin A consumption through the intake of vitamin A-biofortified foods led to increased β-carotene and had a moderate effect on vitamin A status, as measured by serum retinol [97,98].

OFSP offers one of the best sources of naturally-bioavailable β-carotene [95]. This was demonstrated in a randomised, control study conducted in South Africa with primary school children, over a period of 53 school days [66]. Children aged between 5–10 years were randomly selected and divided into two groups. The treatment group consumed 125 g of boiled and mashed OFSP, whereas the control group consumed an equal amount of white-fleshed sweet potato (WFSP) that was devoid of β-carotene [66]. Findings revealed that the OFSP increased the vitamin A status of the children significantly, compared to the vitamin status of children fed the WFSP [66].

A randomised control trial conducted in Zambia established that β-carotene from biofortified maize significantly improved the total body reserves of vitamin A in rural Zambian children. In addition, this could avoid the potential for hypervitaminosis A, which was observed when preformed vitamin A from supplementation and fortification was used [99]. A home-gardening programme, integrated with a primary health care activity with an emphasis on nutrition education, focusing on yellow and dark-green leafy vegetable and OFSP production, was conducted at the community level. Findings demonstrated significantly that the vitamin A status of 2–5 year old children in South Africa was significantly improved [32].

Cassava holds great promise for provitamin A biofortification [100]. However, limited studies have been conducted to evaluate its bioavailability and efficacy [22]. A randomised control trial was conducted in Kenya with children aged 5–13 years old. Although the trial only demonstrated a small improvement in the vitamin A status of children fed provitamin A-biofortified cassava (test group), this was significant compared to children who were fed non-biofortified cassava (control group) [101]. There is an absence of studies on the bioavailability of cassava in South Africa due to cassava not being a staple food in the country.

### 6.2. Iron Bioavailability

The effect of the consumption of iron biofortified foods, like the common bean and pearl millet, has been established. A randomised control trial conducted in Rwanda, among university women that were iron deficient, revealed a significant increase in haemoglobin and total body iron after consuming biofortified common beans, compared to the control group (who consumed non-biofortified common beans) for 128 days [102]. Similar findings on iron status were made in India with biofortified pearl millet. Serum ferritin and total body iron was increased significantly in iron-deficient adolescent Indian boys and girls after the consumption of biofortified pearl millet flatbread twice daily for four months [103]. Findings revealed that children who were fed iron-biofortified pearl millet had a very low prevalence of iron deficiency. Iron deficient children at the baseline were (64%) more likely to significantly reduce or completely overcome their deficiency within six months, compared to the control group [103].

### 6.3. Zinc Bioavailability

Dietary reference intakes for Zn are not met by most South African population groups, regardless of age [15,104]. Zn in biofortified wheat is significantly more bioavailable than Zn in non-biofortified wheat according to findings from various studies [105]. The biofortification of wheat with Zn is not performed in South Africa, probably because it is not grown widely due to the generally unsuitable climate. However, biofortification of the common bean with Zn and iron is under investigation in South Africa [80]. If biofortification of common bean is adopted in South Africa, it is essential to note that bioavailability studies on Zn will have to be interpreted with caution, because, in contrast to iron, Zn has no specific body stores, since it is distributed throughout the body and, as such, is not as readily detectable as iron [106]. This causes challenges regarding the detection and diagnosis of Zn deficiency by Zn concentrations in plasma or serum and other tissues. In addition, the common bean is known to contain significant levels of phytic acid, an anti-nutrient that chelates Zn and iron, reducing their bioavailability [107].

Generally, it is also vital to be cognisant of the fact that bioavailability studies of biofortified foods should be interpreted with caution, because several factors affect micronutrient bioavailability. These include the nutrition and health status of individuals, and anti-nutrient factors in diet and food processing/preparation methods [108,109].

## 7. Retention of Target Nutrients during Processing/Preparation of Biofortified Foods

For the biofortification strategy to be successful, there is a requirement for significant retention of the target nutrients during the handling, processing and storage of biofortified foods. It is hypothesised that, during the processing of biofortified foods, particularly by cooking, the target nutrients may be lost due to chemical changes and physical processes (for example, the leaching of soluble nutrients into water) [110]. Numerous studies have been carried out to explore the retention levels of target nutrients during the processing of biofortified foods, some of which are reviewed next.

### 7.1. Provitamin A Retention

Provitamin A carotenoid retention is affected by different food processing and preparation methods [108,109,110]. Several studies have shown that provitamin A retention is influenced by crop genotype, processing method, recipe and the provitamin A content of the unprocessed biofortified food [110,111,112,113].

Numerous studies conducted in South Africa have evaluated provitamin A carotenoid retention using the traditional methods for the preparation of biofortified foods. In a South African study, Pillay et al. (2014) assessed provitamin A retention during the processing of popular maize foods consumed in KwaZulu-Natal, South Africa. Higher retention of provitamin A was yielded as a result of milling biofortified maize into mealie meal (maize flour), as compared with samp (course cracked/broken kernels [110]. The highest retention of provitamin A carotenoids was observed in cooked *phutu* (thick traditional maize porridge) and cooked *samp*, while preparing the biofortified maize into runny porridge yielded the lowest retention of provitamin A carotenoids [110].

β-carotene retention in the processed OFSP was over 90% when served to South African children in a mashed and boiled form [114]. In Uganda, the retention of all-trans-β-carotene was 78% when the OFSP was boiled in water for 20 min, 77% when OFSP was steamed for 30 min and 78% when the OFSP was deep-fried for 10 min [111]. In Kenya, boiling OFSP appeared to yield a higher true retention of all-trans-β-carotene compared to roasting, and the retention of all-trans-β-carotene appeared to be dependent on the variety of OFSP [115]. However, the process of drying chips caused all-trans-β-carotene content to be significantly reduced; there was a further reduction of approximately 21% when flour was produced from the chips. This study recommended that the OFSP varieties should be consumed boiled to maximise provitamin A intake, as boiling caused relatively lower losses than the other processing methods investigated [115].

Provitamin A retention was assessed in 10 genotypes of OFSP, each characterised at varying intensities, and four different processing methods, which included oven drying, boiling, sun-drying and frying [116]. Provitamin A retention fluctuated according to the method of processing: oven drying produced the highest retention (total carotenoids 90–91% and β-carotene 89–96%) followed by boiling (total carotenoids 85–90% and β-carotene 84–90%) and frying (total carotenoids 77–85% and β-carotene 72–86%). The sun-drying method yielded the lowest retention of total carotenoids (63–73%) and β-carotene (63–73%) [116].

The findings of South African retention studies are consistent with other studies, which indicated that provitamin A retention was affected by processing/preparation conditions [108]. The processing methods, ranked in increasing order for provitamin A retention, were boiling/steaming (80–90%), roasting or frying (70–80%) and sun/solar drying (60–80%) [108], and are in agreement with Vimala and colleagues [116]. With over 80% of beta-carotene in OFSP being retained when boiled, few provitamin A-rich plant foods can match this level [117,118].

### 7.2. Zinc and Iron Retention

A study determined the iron and Zn content in the raw, cooked bean grains (either by pot cooking or pressure cooking). Overall, irrespective of the technique used for cooking, with or without pre-soaking in water, the highest Zn retention was found in the cooked bean grains. However, pressure cooking and pre-soaking in water reduced iron retention in the cooked grains [119,120].

## 8. Consumer Acceptability of Biofortified Foods

One of the indicators of whether a biofortified food will be consumed or utilised for other purposes is its acceptability by potential consumers [121,122]. The availability of biofortified food in gardens or markets does not guarantee acceptability [123]. Some studies on the acceptability of biofortified foods are reviewed in the next section.

### 8.1. Consumer Acceptability

Sensory parameters like colour, taste, smell and texture are influential in the acceptability of foods by consumers [124]. A study conducted to determine the acceptance and preference of provitamin A-biofortified maize in KwaZulu-Natal, South Africa revealed that preschool children significantly preferred biofortified yellow maize to non-biofortified white maize food products (81% vs. 19%) [125]. However, there was no significant difference regarding preference for white and yellow maize by primary school children. In contrast, non-biofortified white maize was preferred over biofortified yellow maize by secondary school and adult subjects [125]. The study suggested that the adults preferred white maize to yellow biofortified maize because they were much more accustomed to the white maize than the younger consumers. For example, the older consumers rated the taste, aroma and colour of the biofortified maize much lower than the ratings given to the same sensory attributes by the children in the sensory evaluation questionnaires. The adults indicated that they did not expect the yellow colour in the maize grain, and that the flavour and aroma of the yellow maize were too strong and unfamiliar. However, younger consumers did not make such comments [125]. Furthermore, focus group discussions conducted in the same study revealed that the lower acceptability of the yellow biofortified maize to adults was due to the stigma attached to yellow maize, as it had been used for food aid and animal feed. Another significant finding of the study was that food type had an influence on the acceptability of the biofortified maize [125]. The researchers suggested that the findings highlighted an opportunity to improve the acceptability of biofortified maize through the selection and/or development of suitable recipes/formulations for provitamin A-biofortified maize.

There is a high risk of VAD during complementary feeding (a period when the new-born baby is 6 to 24 months of age and other nutritious foods have to be given in addition to breastmilk to meet increased requirements for growth and development) in South Africa [40]. A study conducted by [126] explored the acceptance of a complimentary composite food prepared with provitamin A-biofortified maize and chicken stew by caregivers in rural KwaZulu-Natal, and discovered that the acceptance of the complementary foods comprising the biofortified maize was comparable to the control (white maize). Subjects were positive about the taste, texture, aroma and colour of the composite complementary food made from the two varieties of biofortified maize, implying that provitamin A-biofortified maize has the capability to substitute white maize in a complementary diet [126].

Furthermore, Pillay and colleagues assessed the acceptance of OFSP compared to white-fleshed sweet potato (WFSP) as complementary foods by infant caregivers [127]. There were no significant differences concerning the sensory attribute ratings of complementary foods made from WFSP and OFSP among child caregivers. However, the OFSP complementary food was well-accepted, due to its colour and soft texture. This study established that OFSP has the potential to be used in complementary feeding to improve the vitamin A status of infants [127]. Additionally, a study determined that consumers accepted products derived from βcarotene-rich OFSP, including chips, doughnuts, juice and sweet potato leaves [128]. Consumer acceptability was determined by conducting studies at six sites in South Africa’s Limpopo, Mpumalanga, North West and KwaZulu-Natal provinces. Consumer acceptability of the product varied between 85 and 95%, with the highest acceptability observed with doughnuts [128]. Findings indicated that 92% of the consumers preferred the colour of the aforementioned products, 87% of the consumers stated that they would purchase the products and 88% indicated that they would cook the products [128]. However, these opinions differed according to age groups, with younger respondents being unwilling to purchase and prepare these products, compared to the older consumers [128].

### 8.2. Nutrition Education

Community nutrition education has been recognised as a critical strategy that has an influence on the acceptability of biofortified crops and foods [129]. Educating communities regarding nutrition is an essential tool in promoting the nutritional dietary and health benefits of biofortified crops. The acceptability of OFSP in Uganda was achieved partly through the creation of demand and conveying nutrition messages that explained how these OFSP varieties were a good source of vitamin A. Once the mothers had been educated on the importance of vitamin A, they readily adopted the biofortified crop [130]. A study by Khumalo (2011) showed that, although South African consumers preferred commercial white fortified maize meal, contrary to popular belief, they were also willing to accept yellow maize meal. This was mainly for its nutritional value, the knowledge of which the target consumers had acquired from nutrition education [131].

Nutritional education at community level can lead to the improvement of dietary diversity in children ranging between 6–23 months [132]. However, to promote and support the consumer acceptance and adoption of biofortified foods effectively, it is critical to consider how nutrition education messages are generated [123]. It is also essential to recognise the suitable target groups for acceptability tests and nutrition education on biofortified foods. Generally, child caregivers and children are good targets, since these products can also be marketed as improved complementary foods [110].

To ensure the acceptance and effective adoption of biofortified crops among vulnerable populations, promotional activities are essential. Projects aimed at promoting biofortified crops need to utilise intensive nutrition education and extension activities [133]. The important platforms that can be used to increase the adoption and awareness of biofortified crops, while targeting nutritionally vulnerable populations, include nutrition educational programmes held at health clinics and in homes [134]. This would involve discussing nutrition messages in the form of health talks at clinics, which are vital in reaching target populations, especially at ante-and post-natal clinics. The cooking demonstrations performed in the homes of targeted populations will also be effective in illustrating how to incorporate biofortified crops into baby foods and local foods for household consumption [22]. Community level nutritional fairs and talk shows in local vernacular radio stations which discuss maternal and infant nutrition are also very useful in promoting the adoption of biofortified crops among target populations.

Findings from [135] suggest that National Foods representatives state that piloting biofortified maize products in small 2 kg quantities would be an effective strategy to test consumer interest and create awareness of biofortified foods amongst target markets. In addition, building demand within the market for these products through social marketing, focused on target groups such as farmers and vulnerable populations, is essential. This is particularly relevant within the context of gender-sensitive extension guidance and institutional feeding programs, such as school feeding schemes and soup kitchen programmes.

### 8.3. Cost of Biofortified Foods

Food can either be accessed from personal gardens or purchased from the market. The majority of South Africans who do not own small gardens access their foods from the market, especially commercial supermarkets. The adoption of a non-traditional (“new”) food is also affected by its economic accessibility (affordability). In South Africa, child caregivers were willing to buy the biofortified OFSP for use during complementary feeding if it was available and cheaper than the WFSP [127]. Consumer willingness to purchase yellow and commercially fortified maize was contrasted in experimental auctions in three regions in Kenya. The number of consumers most eager to pay for fortified maize (24%) were higher than the discount they required to buy yellow (biofortified) maize (11%) [136]. Existing evidence suggests that biofortified foods are expected to be a cheaper alternative to public health intervention to decrease micronutrient deficiencies in developing countries across the globe [137].

## 9. Conclusions and Recommendations

In South Africa, biofortification has contributed to the availability of provitamin A-rich foods, mainly OFSP, while the biofortification of crops with Fe and Zn is still under investigation. However, it is unfortunate that the prevalence of vitamin A deficiency (VAD) remains unacceptably high in the country, despite the successful biofortification of sweet potatoes. This could be due to the low acceptability of OFSP by adults, who generally make choices at the household level.

In South Africa, research on the biofortification of the common bean with Zn and Fe is underway. If the biofortification is successful and adopted, there would still be the challenge of reducing the levels of anti-nutrient factors, especially phytic acid, to improve the bioavailability of Zn and Fe.

Most South African soils are deficient in nutrients such as Zn, and experience stressful conditions, including drought, which may hinder the achievement of the desired concentrations of target nutrients in biofortified crops. This suggests that the biofortification of crops by traditional breeding and rDNA technology would be useful in increasing micronutrient concentrations in staple crops, but, alone, may not deliver adequate amounts of quality micronutrients and, hence, should be complemented with agronomic biofortification and management.

Research conducted in South Africa suggests that provitamin A-biofortified foods are more acceptable to preschool children than adults, seemingly because adults are much more accustomed to the corresponding traditional (non-biofortified) foods. The improvement of the acceptability of provitamin A-biofortified foods is essential. Nutrition education and the selection and/or development of suitable recipes/formulations for biofortified foods are some of the possible strategies that could be implemented.

Several clinical trials conducted thus far suggest that the bioavailability and efficacy of provitamin A carotenoids in biofortified foods are satisfactory. However, the impact of other factors known to influence nutrient bioavailability, such as processing/preparation methods, dietary fat (enhance absorption of β-carotene), anti-nutritional factors, and the nutrition and health status of the study subjects, should be investigated. Finally, it is essential to note that the strategy of the biofortification of staple crops is beneficial, but, if applied alone, it would not effectively combat micronutrient deficiencies. It should be used to complement other strategies already implemented in South Africa, which include commercial fortification, supplementation and dietary diversity.

For the biofortification programme to reach its full potential, policy makers should recognise the importance of the role of agriculture in the improvement of health. Moreover, national governments and multilateral institutes have to prioritise biofortification in the nutrition agenda. Lastly, food processors and key stakeholders within the value chain must include biofortified crops in their product base. This will ensure that there is more demand and acceptability of biofortified foods, which will enable the attainment of the goal of reaching one billion people by the year 2030.

## Figures and Tables

**Figure 1 foods-09-00815-f001:**
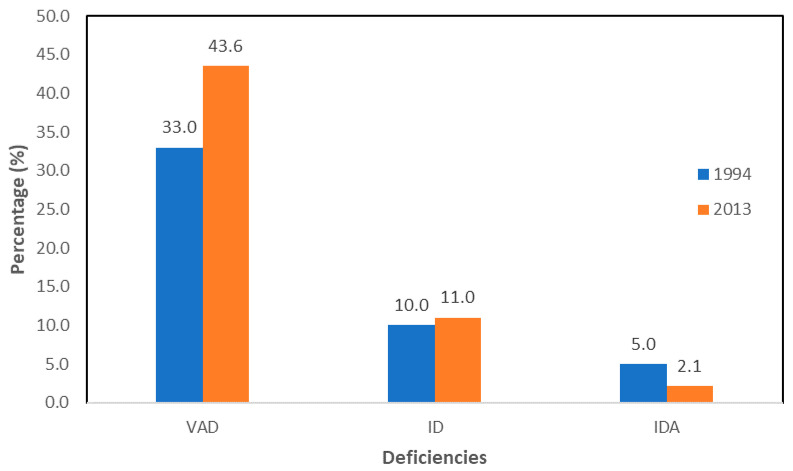
Vitamin A deficiency (VAD), iron deficiency (ID) and iron deficiency anaemia (IDA) in South Africa for 1994 and 2013, for children under five [25,26].

**Figure 2 foods-09-00815-f002:**
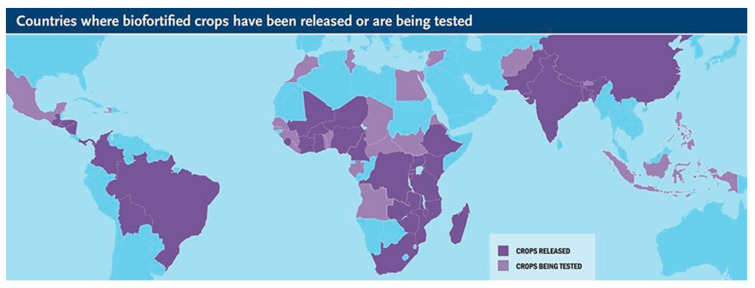
Countries where biofortified crops have been released and are in testing for release [22].

**Figure 3 foods-09-00815-f003:**
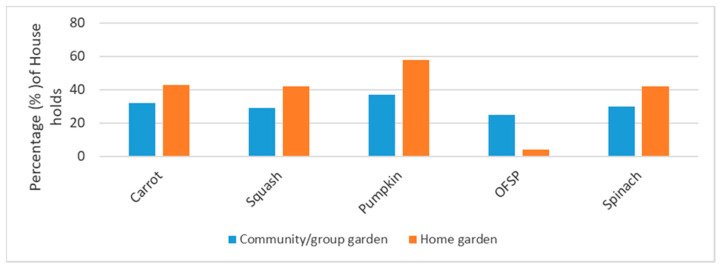
Sources of provitamin A-rich foods for households in the Ndunakazi community, rural South Africa [22].

**Table 2 foods-09-00815-t002:** Staple foods recognized as vehicles for the biofortification of specific micronutrients, agronomic traits and target countries [22,47].

Targeted Micronutrient	Staple Crop	Targeted Country	Agronomic Traits
Vitamin A	OFSP	South Africa, Uganda and Mozambique	Disease resistance, drought tolerance, acid soil tolerance
Vitamin A	Maize	Nigeria and Zambia	Disease resistance and drought tolerance
Vitamin A	Cassava	DRC and Nigeria	Disease resistance
Iron	CB	DRC and Rwanda	Virus resistance, heat and drought tolerance
Iron	Pearl millet	India	Mild dew resistance, drought resistance
Zn	Wheat	India and Pakistan	Disease and lodging resistance
Zn	Rice	Bangladesh and India	Disease and pest resistance Cold and submergence tolerance

OFSP: Orange-fleshed sweet potato; CB: Common bean; DRC: Democratic Republic of Congo.

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
