# Peer review of "Biofortified Crops for Combating Hidden Hunger in South Africa: Availability, Acceptability, Micronutrient Retention and Bioavailability"

_foods, 2020, doi:10.3390/foods9060815_

Round 1
Reviewer 1 Report
The topic is interesting, and I find pleasure to read the manuscript. However, there are few issues and corrections that must be addressed.
Line 90: Reproduce the table rather than copying and pasting from the cited article (because the image is not clear). In addition, indication of table 1 is missing in the text.
Line 161-163: Reorganize the sentence
Line 273: Citation of figure 3
Line 316: Citation?
Line 561: Could you add a paragraph regarding the promotional activities (such as advertisement, campaign, media coverage) including community nutrition education?
Line 642: Web-link is needed including access date
Line 939: Consistency in reference?
Overall comments:
Discuss the methods properly, the way relevant articles were searched using keywords.
Author Response
Reviewer 1
- Line 90: Reproduce the table rather than copying and pasting from the cited article (because the image is not clear). In addition, indication of table 1 is missing in the text.
- Response: This is noted, a table has been reproduced (see line 97) and referred to in line 38.
- Line 161-163: Reorganize the sentence
- Response: This sentence has been reorganized, please see line 167.
- Line 273: Citation of figure 3
- Response: This is noted, a citation has been added for figure 3.
- Line 316: Citation?
- Response: This is noted, a citation has been added, please refer to line 319
- Line 561: Could you add a paragraph regarding the promotional activities (such as advertisement, campaign, media coverage) including community nutrition education?
- Response: This addition has been made to address the promotional activities of biofortified crops and community nutrition education. Please see in text from line 582-600.
- Line 642: Web-link is needed including access date
- Response: The web-link and access date have been added. Please refer to line 665.
- Line 939: Consistency in reference?
- Response: The consistency for the reference has been addressed, see line 976.
- Discuss the methods properly, the way relevant articles were searched using keywords.
- Response: The methods have been discussed adequately, clearly stating the keywords that were utelised to search for relevant articles. Please refer to line 84-91.

Reviewer 2 Report
The authors might want to summarize "Biofortified crops for combating hidden hunger in South Africa". This manuscript would be improved by discussing more biofortification crops via recombinant DNA (rDNA) technology, i.e. vitamin or mineral element biofortification in maize and cassava, etc.
Minor comments:
1) Table 1 quality is too poor.
2) Unit should be same for serum retinol <20 μg/dl, and for serum ferritin < 15 ng/mL, etc.
3) Line 243, 11210 and 9940 μg/100?
4) Line 244, 14210 to 20779 μg/100 g (100 g Dry Weight or Fresh Weight?) should be 142.10 to 207.79 μg/g Dry Weight or Fresh Weight
Author Response
Reviewer 2
- The authors might want to summarize "Biofortified crops for combating hidden hunger in South Africa". This manuscript would be improved by discussing more biofortification crops via recombinant DNA (rDNA) technology, i.e. vitamin or mineral element biofortification in maize and cassava, etc.
- Response: We appreciate reviewer’s comments, the biofortification strategies in most developing countries have been applied without full scientific background hence this review will also assist in understanding the future direction which will also include recombinant DNA (rDNA) technology in most of the crops and postharvest approaches. There has been a stigma of GMO’s which has been topical issue from government to government in the Southern region hence this review provided guidance of significant of biofortification of food particularly in the South Africa as well.
- Table 1 quality is too poor.
- Response: A clear table has been reproduced, please refer to line 97.
- Unit should be same for serum retinol <20 μg/dl, and for serum ferritin < 15 ng/mL, etc.
- Response: < 15 ng/mL was converted to 1.5 μg/dl please see line 135.
- Line 243, 11210 and 9940 μg/100?
- Response: This has been addressed and the correct values have been added i.e. 112.10 and 99.40 μg/g. Refer to line 248.
- Line 244, 14210 to 20779 μg/100 g (100 g Dry Weight or Fresh Weight?) should be 142.10 to 207.79 μg/g Dry Weight or Fresh Weight.
- Response: This was corrected in text please see line 248-249.

Reviewer 3 Report
The theme of the paper is interesting and importante but the work is little discussed and does not allow us to reach important conclusions
The paper aimed to review the prevalence of micronutrient deficiencies (vitamin A, iron, and Zn) in South Africa, but uses dates until 2013 when we are in 2020.
Author Response
Reviewer 3
- The theme of the paper is interesting and important, but the work is little discussed and does not allow us to reach important conclusions.
- Response: We acknowledge the comment. There has not been baseline on how biofortification of foods/feeds in South Africa with the exception that the practice was driven from pollical side rather pure scientific bases to allude synergetic responses of combining foods. This review will build scientific theoretical responses to biofortification strategy in Southern Africa.
- The paper aimed to review the prevalence of micronutrient deficiencies (vitamin A, iron, and Zn) in South Africa, but uses dates until 2013 when we are in 2020.
- Response: The review aims to preclude what has been done in practice hence nothing has been done to scale up biofortification from scientific perspectives. Therefore, to dig deep from literature germane was fruitful to understand what has been done. The trials are currently underway which will produce relevant scientific rationale of biofortification with different horticultural and agronomic crops.

Round 2
Reviewer 2 Report
The authors might want to summarize "Biofortified crops for combating hidden hunger in South Africa". This manuscript would be improved by discussing more biofortification crops via recombinant DNA (rDNA) technology, i.e. vitamin or mineral element biofortification in maize and cassava, etc.
- Response: We appreciate reviewer’s comments, the biofortification strategies in most developing countries have been applied without full scientific background hence this review will also assist in understanding the future direction which will also include recombinant DNA (rDNA) technology in most of the crops and postharvest approaches. There has been a stigma of GMO’s which has been topical issue from government to government in the Southern region hence this review provided guidance of significant of biofortification of food particularly in the South Africa as well.
My further comment: If the authors can compare the vatamin or Fe or Zn corns and cassava etc. derived from transgenic plants and conventional breeding it will make the manuscript much stronger.
Author Response
We acknowledge and appreciate the comments made by the Reviewer. We have attached our written responses.
